# Explaining rising caesarean section rates in urban Nepal: A mixed-methods study

**Sulochana Dhakal Rai**[ID][1]*, **Edwin van Teijlingen**[1], **Pramod R. Regmi**[1], **Juliet Wood**[1], **Ganesh Dangal**[ID][1,2], **Keshar Bahadur Dhakal**[3]

1 Bournemouth University, Bournemouth, United Kingdom, 2 Kathmandu Model Hospital, Kathmandu, Nepal 3 Karnali Province Hospital, Surkhet, Nepal

* s5085651@bournemouth.ac.uk, dhakal_sulochana@yahoo.com

## Abstract

### Introduction

Caesarean section (CS) rates are rising in urban hospitals in Nepal. However, the reasons behind these rising rates are poorly understood. Therefore, this study explores factors contributing to rising CS rates in two urban hospitals as well as strategies to make rational use of CS.

### Methods

This cross-sectional mixed-methods study was conducted in 2021 in two hospitals, one public hospital and one private one in Kathmandu. The quantitative part included a record-based study of 661 births (private hospital = 276 and public hospital = 385) for the fiscal year 2018/19. The qualitative part included semi-structured interviews with 14 health professionals (doctors, nurses & midwives) and five key informants from relevant organisations and four focus group discussions with pregnant women in antenatal clinics in two hospitals. Quantitative data were analysed using SPSS v28. Qualitative data were organised through NVivo v12 and thematically analysed.

### Results

The overall CS rate was high (50.2%). The CS rate in the private hospital was almost double than that in the public hospital (68.5% vs. 37.1%). Previous CS was the leading indication for performing CS. Non-medical indications were maternal request (2.7%) and CS for non-specified reasons (5.7%). The odds of CS were significantly higher in the private hospital; women aged 25 years and above; having four or more antenatal clinic visits; breech presentation; urban residency; high caste; gestational age 37-40 weeks; spontaneous labour and no labour. Robson group 5 (13.9%) was the largest contributor to overall CS rate, followed by group 1 (13.4%), 2 (8.8%), 3 (4.4%) and 6 (2.9%). Similarly, the risk of undergoing CS was high in Robson groups 2, 5, 6, 7 and 9. The qualitative analysis yielded five key themes affecting rising rates: (1) medical factors (repeated CS, complicated referral cases and breech presentation); (2) socio-demographic factors (advanced age mother, precious baby and defensive CS); (3) financial factors (income for private

**Data availability statement:** All relevant data are within the paper and its Supporting information files.

**Funding:** The author(s) received no specific funding for this work.

**Competing interests:** The authors have declared that no competing interests exist.

hospitals); (4) non-medical factors (maternal request); and (5) health service-related factors (lack of awareness/midwives/resources, urban centralised health facilities and lack of appropriate policies and protocols). Four main strategies were identified to stem the rise of CS rates: (1) provide adequate resources to support care in labour and birth (midwives/trained staff & birthing centres); (2) raise awareness on risks and benefits mode of childbirth (antenatal education/counselling and public awareness); (3) reform CS policies/protocols; and (4) promote physiological birth.

## Conclusion

The high CS rate in the private hospital reflects the medicalisation of childbirth, a public health issue which needs to be urgently addressed for the health benefits of both mother and baby. Multiple factors affecting rising CS rates were identified in urban hospitals. This study provides insights into factors affecting the rising CS rate and suggests that multiple strategies are required to stem the rise of CS rates and to make rational use of CS in urban hospitals.

## 1. Introduction

Caesarean section (CS) is a lifesaving major surgical intervention in high-risk pregnancies to save mother and/or foetus, and the use of CS evolved throughout history [1]. In 1985, the World Health Organization (WHO) recommended a CS rate of 10-15% at population level [2]. More recently, it advocated that CS is only used when there are medical indications [3], since CS rates higher than 10% at population level do not result in lower maternal and new-born mortality rates [4,5]. However, CS rates are rising worldwide, although with regional and national disparity. Globally, rates in 2000 were double (21.1% vs 12.1%) that in 2015 [6]. It is predicted that the global CS rate will increase to 28.5% in 2030 with the lowest rate (7.1%) in Sub-Saharan Africa and highest (63.4%) in Eastern Asia [7]. There are many health consequences of CS to mother (increased mortality risk, severe morbidity and a higher risk for adverse outcomes in subsequent pregnancies) and child (altered immune development, allergy, atopy, asthma, and reduced diversity of gut microbiome) [8]. Moreover, CS is a financial burden for individuals, their family, and the health system/country [9].

The rates of CS are increasing rapidly in South Asia [10], as they are in Nepal [10,11]. The higher CS rates in urban and private hospitals are linked to "Too Much Too Soon" and the medicalisation of pregnancy and childbirth [11]. Although there is a disparity in rates between urban and rural, the overall rate of CS rose three-fold from 1996 to 2016 in Nepal. In private hospitals it rose from 8.9% in 1996 to 26.3% in 2016 [12]. However, the reasons behind the rising rates are not well understood. Hence, this study explores factors associated with rising rates of CS in two urban hospitals in Nepal. Specifically, this study has estimated the CS rate in study samples using the Robson groups; identified key factors contributing to a higher rate of CS and sought strategies to improve the rational use of CS.

## 2. Methods

This mixed-methods cross-sectional study was conducted in 2021 in one public hospital: Paropakar Maternity & Women's Hospital (PMWH) and one private one: Kathmandu Model Hospital (KMH). Both qualitative and quantitative data were collected concurrently, but quantitative and qualitative data were analysed and presented separately. Data integration was performed in the discussion [13], weaving qualitative and quantitative findings together by themes [14].

The hospital record-based study included 661 births (public n = 385; private n = 276) for the fiscal year 16/07/2018–15/07/ 2019. Retrospective data were collected from 1st August to 30th October 2021 from the fiscal year (16/07/2018–15/07/2019). Samples were selected from the hospital records (discharge registers/patient records) using systematic random sampling. The sample size for the public hospital was calculated using the formula: $n = z^2p \, (1\text{-}p)/d^2$. However, for the private hospital it was calculated using the formula: 5% margin of error, 95% confidence level and representing 50% of sample proportion. An additional 10% sample was collected compensating poor/incomplete records. Selected data from hospital medical records were 4compiled in a standardised format.

The qualitative aspect of this study included interviews with 14 health professionals (seven doctors, five nurses, two midwives) working in maternity departments of the selected hospitals and five key informants from relevant organizations (two hospital directors & one representative each from Nepal Society of Obstetrics and Gynaecologists [NESOG]; Midwifery Society of Nepal [MidSoN]; Ministry of Health & Population Nepal [MoHP]). Similarly, four focus group discussions (FGDs), two in each hospital, were conducted with a total of 22 pregnant women (4-6 participants each FGD) at antenatal clinics. Purposive sampling was applied for both interviews and FGDs. Interviews and FGDs were conducted from 1st August 2021 to 30th October 2021 concurrently with quantitative data collection. Interview guidelines were used to facilitate interviews and FGDs. The interview and FGD tools primarily focused on the reasons for conducting and requesting CS, decision-making regarding CS, barriers and challenges to reducing its use, and strategies for promoting the rational use of CS. These tools were pretested with health workers and pregnant women in Karnali Province Hospital in Nepal in late 2019, minor modifications were made to both tools following the pilot.

Quantitative data were analysed using SPSS version 28. Statistical significance is set at $p < 0.05$. Descriptive analysis of study variables, Chi-square test, logistic regressions and risk ratio are performed. The sample data were classified according to Robson 10 groups classification, based on parity and previous CSs, number of foetuses, gestational age, foetal presentation and lie, and onset of labour [15]. The contribution of each group to overall CS rate were analysed and identified. For each Robson group risk ratio (RR) with 95% confidence interval for birth by CS is calculated to estimate risk or incidence of CS in each Robson group. Qualitative data were organised using NVivo 12 and analysed thematically [16,17]. Both qualitative and quantitative findings integrated into relevant themes or concepts through a narrative technique using weaving approach [14].

Ethical approval was obtained from the ethical review board of Bournemouth University (Ref: 26403) and Nepal Health Research Council (Ref: 3503). Additionally, approvals were granted by Institutional Review Committees of both study sites (PMWH, Ref: 59-11-2214 & KMH, Ref: 089-076). A written consent was obtained from both interviewees and FGD participants.

## 3. Results

### 3.1. Rates of CS

The overall CS rate was very high (50.2%), and almost double in the private hospital (68.5% vs 37.1%). Overall, there were more elective (55.7%) than emergency CSs (44.3%), whilst the elective CS rate (61.9%) was higher than the emergency rate (38.1%) in the private hospital. However, in the public hospital, the elective CS rate (47.6%) was lower than the emergency CS rate (52.4%). The primary CS rate was 71.1%. All women in the study (100%) who had a previous CS had a repeat CS, and the incidence of repeat CS was 28.9%. Similarly, all women

(100%) with a breech presentation ended up with a CS. Socio-demographic characteristics of women are shown in S1 Table.

All interviewees highlighted that the overall CS rate is high, but much higher in private hospitals than public hospitals, as an interviewee from the private sector highlighted:

> *"Our hospital has a CS rate of 60%, although this rate is higher than a public hospital. It is lower than any other private, corporate hospital. But even this 60% is considered very high and it is a matter of concern." (P9)*

### 3.2. Indications for CS

Previous CS (27.4%) was the most common indication for performing CS (S2 Table), it was also the leading indication in the private hospital (18.7%), performing repeat CS was more than twice as high as in the public hospital (18.7% vs 8.7%). Foetal distress was the most common indication for CS in the public hospital (10.5%). Maternal request (2.7%,) and CS for unspecified reasons (5.7%) were the key non-medical indications overall. Interestingly, CS for maternal request was only recorded in the private hospital (4.8%), whilst CS for none specified reasons was evident in both the public (2.7%) and private hospital (3.0%).

Previous CS (27.4%) was the leading indication for elective CS and foetal distress (14.5%) was the principal indication for emergency CS (S3 Table). Maternal request (2.7%) was only recorded for elective CS. However, CS for not specified reasons was higher in emergency CS (3.9%) than in elective CS (1.8%) (S3 Table). The key indication for CS among nullipara women was foetal distress (11.5%) and the major indication for CS in multipara was previous CS (27.4%) followed by foetal distress (3.0%). The proportion of CS for maternal request and not specified reasons were higher in nullipara than multipara (S4 Table).

Interviewees also thought that CS is performed mainly for medical reasons:

> *"The main indications for caesarean section are medical indications, in medical indications, sometimes there is low maternal height, CPD, sometimes low progression of labour, sometimes placenta previa, even if there is previous CS then caesarean section is done here…" (K3)*

### 3.3. Decision making around CS

All interviewees said that decisions around CS are made by senior doctors, based on indications of CS, and women and their family are consulted:

> *"… the decision to conduct or not to conduct CS is done by a consultant. We also discuss it with the client and her family members, and we decide." (P10)*

All interviewees said that decision making around elective CS is usually pre-planned before admission to labour ward. However, decision making around emergency CS is made if complications occur during labour:

> *"Elective CS are usually planned…and the patients are counselled for the CS informing their indication. Decision for emergency CS is made in the following condition like if there is foetal distress. If there is danger for both mother and baby, less foetal movement, or any complications arise during delivery like excessive blood loss, antepartum haemorrhage." (P9)*

All interviewees agreed that women and their families are informed in case of an emergency CS, whilst women and their family are already informed before admission in case of elective CS. Consent is taken before each CS:

*"Pregnant women and families are included in the decision before doing CS. Because without their consent, the decision cannot be made by the doctor alone. At the time of elective, the patient and their family already know… In the emergency case, due to the urgent need to do CS, the patient or family decides to sign the consent form after explaining everything to the people who have come with patient." (P5)*

### 3.4. Robson Classification of CS

Robson group 1 was the largest (33.1%), followed by group 3 (20%), 5 (13.9%) and 2 (13.8%). These four groups contained 80.8% of the total sample. Similarly, CS rates were high in low-risk groups (1-4). In addition, Robson group 5 (13.9%) was the highest contributor to overall CS rate followed by groups 1 (13.4%), 2 (8.7%) and 3 (4.4%) (S5 Table).

In the public hospital, Robson group 1 (11.9%) was the highest contributor to overall CS rate followed by groups 5 (7.5%), 2 (4.7%), 3 (3.4%) and 6 (3.4%). However, in the private hospital, Robson group 5 (22.8%) was the biggest contributor to the overall CS rate, followed by groups 1 (15.6%) and 2 (14.5%). The lowest contributor was Robson group 9 in public hospital and group 8 in private hospital (S6 Table). The risk ratio for CS was calculated to identify incidence or risk of CS in each Robson group. S7 Table shows that the risk of ending up with a CS was higher in Robson group 5 (RR: 2.37), 6 (RR: 2.05), 7 (RR: 2.02), 9 (RR: 1.99) and 2 (RR:1.33).

The risk of CS was significantly lower in Robson group 3 in both hospitals. Robson group 10 had a significantly higher risk of undergoing CS in public hospital (RR: 1.100). However, the risk of CS was significantly higher in Robson group 1 (RR: 1.18) and 2 (RR: 1.21) in private hospital (S8 Table).

### 3.5. Factors associated with mode of childbirth

Nine (out of twenty) variables were significantly associated with mode of childbirth [S9 Table]. Independent variables, significantly associated with mode of childbirth were further analysed and bivariate logistic regression analysis shows that the chance of delivering a baby by CS is 1.37 times higher in the private than in the public hospital. Mothers aged 25-29 were 1.40 times, 30–34-year-old women were 1.41 times and those 35 years and older were 1.62 times more likely to undergo CS than those 20 or younger. Women in gestational weeks 30-36 were 0.78 times less likely to have a CS compared to women in gestational week 37-40. Women with a breech presentation were 1.69 times more likely to undergo CS compared to those with a cephalic presentation. Women who had spontaneous labour were 1.18 times, and women who had no labour during childbirth, 2.26 times more likely to undergo CS than women who had induced labour. Similarly, women who attended four or more ANC visits had 1.26 times higher probability of undergoing a CS than those who had three or fewer visits. High caste women were 1.14 times more likely to have a CS than middle-caste women. Likewise, urban residents were 1.09 times more likely to have a CS than rural women (S10 Table).

Similarly, the multivariate regression model was also applied to identify the significant factors associated with rising CS. Women in private hospital were 1.32 times more likely to have a CS compared to in the public hospital. The age of the mother was also a factor for CS. Mothers aged 25 to 29 years were 1.17 times and those aged over 35 1.29 times more likely to undergo CS compared to the mothers aged less than 20 years. Mothers in spontaneous labour were 1.16 times and in pre labour were 1.96 times more likely to undergo CS compared to those having induction. Similarly, mothers with four or more times ANC visits were 1.26 times more likely to undergo CS compared to those who attended fewer than four visits (S10 Table).

### 3.6. Key factors affecting the rise in CS

Five key themes including several sub-themes were identified behind the rising CS rates: (1) medical factors; (2) sociodemographic factors; (3) financial factors; (4) non-medical factors; and (5) health service-related factors (S1 Fig).

**3.6.1. Medical factors.** Medical factors included three sub-themes: (1) repeat CS and lack of VBAC (vaginal birth after CS); (2) complicated referral cases; and (3) breech presentation (S1 Fig).

Most interviewees said that repeat CS and lack of VBAC were the major factors for rising rates, because a repeat elective repeat CS is planned for women with a previous CS due lack of intensive monitoring as well as to avoid risk. Hospitals do not want to take the risk, but neither it seems, do the women and their families:

> *"We don't provide the opportunity of VBAC because it needs intensive monitoring of patient as well as hospital do not want to take risk. Previous CS falls under indication plus patient also demand it, so we do repeat CS for previous CS." (P9)*

Another main medical factor was the complexity of cases referred from smaller units/hospitals elsewhere in Nepal. Many highlighted that both hospitals are referral centres, and many complicated cases who need CS were referred to these hospitals. Sometimes complicated cases arrived at urban hospitals from remote area by air ambulance:

> *"… the reason for the high caesarean section rate in this hospital is because this is a referral centre also and complications that arise in other places are referred to here through helicopter or nearby places. Since this is a main referral centre that is the reason for the high caesarean section rate." (P2)*

Elective CS for breech presentation is a reason for the rising rate of CS. Trial of labour was not planned for primigravida with breech unless the woman is admitted in active labour with fully dilated cervix and buttock of baby is visible. However, trial of labour is offered to multigravida if their condition is deemed favourable:

> *"Labour trial is not given for primigravida breech for first-time mom, but when they come from home if the dilation is over 10 centimetre and if the buttock of the baby is seen then we do not tell them to perform a caesarean section, but rest of the time on first-time mother receive a caesarean section. But if it is multigravida, and if they have normal condition then we try to perform normal delivery." (P2)*

**3.6.2. Socio-demographic factors.** Three sub-themes were identified: (1) changing socio-demographic characteristics of women (pregnancy in advanced age and career-oriented rather than family); (2) precious baby; and (3) lack of security of service providers, legal issues and defensive CS (S1 Fig).

The socio-demographic characteristics of the obstetric population are changing. This study indicated that pregnancy at advanced age was a reason for the increasing rate of CS. Problems of infertility, diabetes and hypertension also increase with advanced age, and these are associated with CS:

> *"…the women who come to us are slightly older. We need to perform CS on elderly women and with many complications. So, because of this also CS rate is slightly high." (P10)*

Women focus on their education, work, and career rather than having children. They get pregnant at a later age, when subfertility and complications increase the likelihood of CS.

> "…the main reason for the increase in CS is nowadays women are more concerned about their education and career. So, they do late marriage and plan baby late, there are also subfertility cases. All these factors lead to the complications, for which CS is needed." (P12)

Pregnancies after infertility treatment, after many years of marriage and having a poor obstetric history may be viewed as particularly precious. Many interviewees emphasised that such babies are more likely to be delivered by CS:

> "Precious babies are also more likely to be caesarean section. If they have had a baby by IUI (=intrauterine insemination), IVF (in-vitro fertilisation), have had a baby after years, have had a miscarriage, or have had a previous pregnancy loss, have had a bad experience such as stillbirth…they mostly demand CS." (P7)

In Nepal, there is a lack of security and legal protection for doctors/service providers in case of unfortunate incidents. Doctors, nurses, and hospitals are criticised and may even attacked and punished if something goes wrong with the mother or baby due to vaginal birth rather than CS. Many thought that doctors perform defensive CS for self-protection to avoid verbal/physical assault and litigation:

> "… If one baby dies because of conducting vaginal delivery rather than CS, Then, hospital is vandalised, and doctor is charged and court orders to both hospital and doctor. That is why doctors do not want to take risk…" (K4)

**3.6.3. Financial factors.** Two sub-themes occurred under this theme: Income source for private hospitals and incentive for public hospitals (S1 Fig). Many interviewees highlighted that CS was a major operation which can be a good income source for private hospitals.

> "It comes with the financial aspects of a private hospital. Because of these things, I think it is a challenge to conduct normal delivery." (P8)

Financial benefit also includes an increased bed occupancy:

> "…benefit of CS to the hospital is definitely in terms of money. After CS, hospital receives income from bed/room, income from medicine, income from OT (=operating theatre) charge, many benefits from CS... income source to the hospital, staff also get salary on time." (P10)

CS is conducted in private hospitals to generate money as they charge a lot for a CS:

> "…the intention is to earn money in a private hospital. If you do normal delivery, the money will not come. Intentionally, they want to cut women to generate the income… Private hospital charge approximately 1-2 lakh rupees (=625-1,250 US$) for a caesarean section." (K3)

Public hospitals also receive some incentives from government. Some interviewees highlighted that the state provides some funding to hospitals to provide free CS:

> *"…the Ministry of Health provides the service free of cost from the Safe Motherhood programme and the patients do not have to pay money for it. We keep the statistics of how many surgeries we performed and send it to the Ministry of Health and the Ministry of Health provides 8000 Nepali rupees* (= 50US\$) *per caesarean section." (P4)*

The reimbursement for doing CS is higher than for vaginal birth:

> *"Upon doing caesarean section the Government of Nepal provides incentive, which is even more than that provided to normal delivery…" (P2)*

**3.6.4. Non-medical factors.** Two sub-themes emerged here: Maternal request and service providers' attitude towards CS (S1 Fig). All women in FGDs said that the preferred mode for their forthcoming birth is a normal delivery due to the benefits for both mother and baby:

> *"If normal delivery, it is good for both mother and baby because in caesarean section incision is done in abdomen which creates difficulty in sitting, walking. Normal delivery is natural, and caesarean section is difficult, and it is different. So, normal delivery is good for both mother and baby. Natural birth is better than incision in abdomen." (FGD/P22)*

However, CS performed at maternal request is also one of the main non-medical factors for the rising CS rates in urban hospitals in Nepal:

> *"…Some of the CS are done on demand of the pregnant mothers as they do not want to give normal delivery in any condition. So, because of this also CS rate is slightly high." (P10)*

Cord around the foetal neck (nuchal cord) is sometimes evident on ultrasound scans. It is not a standard indication for CS as it does not usually pose a threat to the foetus. If a pregnant woman learns that her baby has its cord around the neck, she may then request a CS to avoid perceived risks to baby. Doctors may consider the autonomy of women and respect their request when the cord is awkwardly positioned:

> *"…major reason is patient request. Even if there is no indication for example cord round neck is not indication for CS but while we explain patient about this, they themselves demand CS. I think caesarean delivery on maternal request is one of the major unnecessary CS, in that case we should also respect patient request, so we have to do it." (P11)*

Some indicated that service providers also perceive CS as the safest option to avoid risk. Normal birth is not promoted, and CS is performed by doctors 'to be on the safe side':

> *"…normal delivery is not encouraged as much as it used to be. In case of CS, baby is safe, mother is also safe which reduces the headache of the doctors."* (P7)

**3.6.5. Health-services-related factors.** Four main sub-themes are identified here: (1) lack of awareness around mode of childbirth/risk and benefits of CS and vaginal birth; (2) lack of adequate resources; (3) the centralisation of health facilities in urban Nepal; and (4) lack of appropriate policies and protocols on CS (S1 Fig). Most of interviewees highlighted that women's lack of awareness around mode of childbirth is one of the main factors for the rising CS rate. Interviewees described how most pregnant women were poorly informed about the risks and benefits of CS in ANC:

*"…there is lack of awareness among the women themselves on the risk and complications of caesarean section, so they want to do it. Whether the caesarean section is required, or not. This type of awareness is lacking among the women as they are not well informed about it. In the hospital settings also, there is not any system to provide information on this while the women come for ANC visit. There is communication gap between the service providers and client. This is one of the main barriers to reduce CS." (K3)*

Furthermore, it is highlighted that health workers are not able to inform pregnant women nor teach them pain relief techniques:

*"…we health workers do not provide full information to them from our side and do not make them aware about the complication of caesarean section and we do not tell them much about the techniques of pain relief. This is not quite practised in Nepal and therefore caesarean section is mostly done."(P2)*

Some interviewees highlighted that urban, educated and rich women are not aware of mode of childbirth:

*"Urban educated women are not aware and informed about the physiological birth. I think, that's why they think that birth is pathological condition. So that, they choose artificial way to bring their baby in this world… Even, women rights activists should be aware because those women who call themselves feminist or empowered women, they mostly do CS while giving birth to baby…" (K3)*

Most interviewees emphasised that a lack of adequate resources, especially midwives or SBAs (skilled birth attendants) is another main factor for the rising CS rates, as some mentioned:

*"Main challenge is proportion of doctor/nurses with proportion of patients, so if trained midwife, SBA trained personnel are mobilised in every remote as well as urban then this may help to reduce unnecessary CS rate."* (P7)

Some interviewees highlighted that midwives who are expert for handling physiological births are lacking in Nepal:

*"…We don't have experts (midwives) in handling natural births in Nepal. Recently our country has produced 28 midwives. It takes a minimum of 10 years for them to adopt a system-adjusted society because it is challenging for you to change the established practice to change the system…" (K3)*

Some also noted the lack of medical skill in conducting vaginal and instrumental:

*"…we called instrumental delivery like forceps, vacuum delivery nowadays their use is very low like when I was MD [medical] student about 25 years ago. At that time there was vacuum delivery 6% and forceps delivery 3% - total 9% of instrumental delivery. Now, this 9% is also added to caesarean section because today's doctors are lacking the skills of instrumental delivery like vacuum, forceps…" (K4)*

The centralisation of comprehensive maternity health facilities in tertiary centres in cities is another reason for rising CS rates in urban Nepal. Many interviewees highlighted that big and/or private hospital are situated in urban areas:

*"The trend of caesarean section in Nepal is increasing now especially in urban areas, in big hospitals of cites, corporate hospitals in comparison to rural and remote areas…" (K1)*

Another key issue is the lack of written protocols and policies on CS, lack of models of maternity care and respectful maternity care, lack of policy on rewards for low CS or monitoring and supervision regarding high rate of CS, lack of political commitment to implementing international recommendations (Robson classification) and lack of appropriate auditing of CS. Most interviewees said there were no written policies and protocols for CS:

*"There is no such a separate strategy/policy for CS in Nepal, we follow MD level book and according to the indication we do CS." (K1)*

Hospitals in Nepal mostly follow an obstetric model of maternity care. Some interviewees importantly stressed that the lack of a midwifery model and current practice of an obstetric model of care is also a reason of increasing the CS rate:

*"…doctors are trained to handle pathological conditions while being professionally trained...where there is a midwife, the midwifery is called the social model of maternity care. Similarly, the midwifery model of maternity care, women are trained to be empowered…if professional midwives are very empowered in the country, the rate of CS is relatively low in that country." (K3)*

One participant indicated that the lack of respectful maternity care was also a reason for unnecessary CSs in Nepal. Performing unnecessary CS is obstetric violence. However, there was no complaints system and service users are scared to raise their voice:

*"...There is lacking respectful maternity care…There is not any mechanism to handle consumer rights although there is legal right. There should be a system to complain, but even clients are very fearful even to raise their voice. Unnecessarily CS is obstetrics violence, which is what is happening in Nepal, first it was in Brazil. I am blaming the system for the mindset of the society which is the big drawback of the system that brought women up." (K3)*

The Government of Nepal has no policy to reward hospitals with lower CS rates or a system for monitoring and supervision to reduce high CS rates. Some interviewees raised the issue about lack of policy and protocol for awards to hospitals for low CS rates and monitoring of CS rate in hospitals:

*"…Government has no policy to award the hospitals having caesarean section rate less than 15%. And another thing is there should be the monitoring and supervision of the hospitals in order to prevent the higher rate of caesarean section..." (P2)*

A lack of political commitment to reduce CS rate was clear according to several interviewees:

*"…national leader, you must be committed to implementing the standards set by the WHO.... from our Prime Minister it is impossible, then in the Ministry of Health, the ministers in the Ministry of Health also come from different backgrounds we cannot expect that from them…... For example, WHO says use the Robson classification, but if the director of the maternity hospital cannot get support from the Government than he/she could*

*not implement it alone, that is what the system is all about we need system to support our work… NESOG doesn't seem like to have taken any initiative to control... (K3)*

Lack of appropriate monitoring and auditing systems can be a reason for rising CS rates. Most said that they do record CS in medical record and presented report on CS along with other maternal health statistics monthly and yearly basis:

*"In our hospital, there is a monthly audit …for example, how many of the deliveries during the month are normal delivery, how many forceps, vacuum deliveries and how many caesarean sections are done, how many are done annually. We also provide this data to the Government of Nepal." (K1)*

However, a few said that they do audit CS according to Robson criteria:

*"…there is a system for audit of CS. We do CS audit according to the Robson criteria." (K2)*

In contrast, some interviewees agreed that they would not do audit of CS and it would be better to do CS audit using Robson criteria:

"*We do not have practice of audit of CS. If we move forward using Robson criteria, then it will be better." (K3)*

## 3.7. Strategies for optimal use of CS

Four main themes including several sub-themes were identified regarding the strategy for the rational use of CS (S2 Fig): (1) adequate resources; (2) raising awareness around mode of childbirth/risk and benefits of CS and vaginal birth; (3) reforming policy and developing protocols for CS; and (4) the promotion of physiological birth.

### 3.7.1. Adequate resources.
Three sub-themes yielded in this theme: (1) training and availability of SBA trained staff, (2) training, education and utilisation of midwives; and (3) the establishment of birthing centres (S2 Fig).

The Government of Nepal should focus on training and mobilising SBAs. All SBAs should be skilled in normal birth, and they should be placed at labour room and delivery room to support women for normal delivery as well as to facilitate birth. Most interviewees emphasised that providing SBA training to doctors and nurses and availability of adequately trained staff would be a strategy for reduction of the CS rate and the promotion of normal birth:

*"Skilled birth attendants have to be mobilised in every part of the country. Likewise, the service providers should be trained and skilful to provide this type of services... Along with doctors, SBAs also should have a hand in normal delivery, so the government should focus on training and mobilising SBA. In the same way there should be skilled birth attendants at labour room, delivery room to perform the normal delivery. At the hospital level we need to influence the patients for the normal delivery as well as staff themselves should be skilful." (P13)*

Midwives should be posted in antenatal clinics, birthing centres, PNC (postnatal care) clinics, as well as in the community. Most interviewees also highlighted that education and appropriate utilisation of midwives must be the strategy for reasonable use of CS and physiological birth:

*"First of all, midwives should be produced, and they have to be mobilised from ANC, birthing to PNC. Because of the lack of midwives normal physiological birth is not promoted. If we are able to produce more midwives and utilise them from community level to higher level than the rate of caesarean section can be decreased." (P2)*

Similarly, midwives who can promote and assist physiological birth should be trained in large numbers in Nepal:

*"…if there is a midwife then they should promote women for the normal physiological birth until and unless there is an emergency otherwise you can have a normal birth, you are made with such a physiology type of counselling should be done among the mothers... Until and unless we don't have a midwife who can assist the physiological birth, it would not happen. Human resources who can handle physiological birth need to be produced in large numbers; the government has already estimated that 9000 midwives should be produced and utilised..." (K3)*

Birthing centres should be established in each hospital to promote physiological birth. Many interviewees stressed the importance of establishing birthing centres in each hospital which are run by midwives and where normal birth is promoted:

*"In order to promote normal physiological birth, birthing centres should be established, well trained nurses should be utilised in every health facility." (K2)*

Furthermore, these birthing centres should be run by midwives

*"Birthing centres should be established which are run by midwives…" (K4)*

**3.7.2. Raising awareness around mode of childbirth.** Three sub-themes were recognised under this theme: (1) counselling of pregnant women for normal birth, (2) increasing awareness of pregnant women around mode of childbirth, (3) increasing public awareness around mode of childbirth (S2 Fig).

Doctors should counsel women to motivate them to have normal births. Most interviewees stressed that counselling pregnant women about normal birth from the beginning of pregnancy can play a vital role in reducing CS rates:

*"…If we have counselled properly during antenatal period, the client will be prepared mentally to give normal birth…Doctors should counsel that you have come to safe place, and you have a normal delivery. Counselling also convinces many patients, so, I think as a doctor he/she should also be a counsellor." (P8)*

Some interviewees also stressed importance of counselling women from their preconception period regarding the process and advantages of normal delivery to reduce fear of normal birth:

*"…women should be counselled from their preconceptions. About how one can deliver normally then only we might be able to reduce their fear of the normal delivery by explaining to them about the benefits of normal birth. This way we might be able to reduce the rate of caesarean section." (P1)*

Many participants emphasised health education to pregnant women on mode of childbirth:

*"Health education should be given to pregnant women by health workers and encouraging normal delivery." (FGD/p1)*

Furthermore, some stressed that childbirth education should be given during ANC visits:

*"Education on mode of childbirth should be given to women during antenatal check-up. And their mental and physical aspect should be taken care of to prepare them. If you can do that then CS will decrease. Woman must be prepared for a normal delivery rather than CS." (P2)*

Some interviewees added that women should be made aware of the benefits of vaginal birth:

*"…make pregnant women aware about the benefits of vaginal delivery during ANC visit. Monetary cost is also cheap, physical recovery is also quick, CS recovery takes time so normal delivery recovers quickly, all these things should be made aware during ANC visits." (P10)*

Many interviewees stressed that pregnant women should be empowered by providing service from midwives to help women make informed decisions around the mode of their childbirth:

*"The first thing is that the women must be empowered themselves, there is not much human resource to empower the right of the woman and that is lacking for example the midwives… the main thing that can be done is the empowerment of the woman to reduce the rate of caesarean section. And the number of people performing elective caesarean section in the urban sector is increasing and women need to be aware that they must be empowered so that they can make decisions for themselves." (P2)*

Some interviewees highlighted that especially educated urban women should be made aware about the risks and benefits of different modes of childbirth – both risks and benefits of CS and normal birth:

*"... All we need to do is to make aware and informed urban settings women about the consequences of CS…Likewise, women should be made aware and informed about the benefits of normal physiological birth." (K3)*

Many interviewees emphasised that public awareness around mode of childbirth should be increased using mass media such as displaying posters and pamphlets:

*"Normal delivery has to be promoted via ANC by displaying posters, pamphlets and by counselling patients in the OPD about the benefits of normal delivery. This can be done by providing information in an understandable form in television." (P3)*

Some also emphasised the importance of educating family and the wider community:

*"…family, community people should understand that CS should only be done in indicated cases...make the client aware from the very beginning to promote normal delivery." (P9)*

Some interviewees recommended increasing public awareness on CS, and how health education around childbirth could occur in high schools:

*"Changes should be done only from education system. Health education should be provided in courses of +2 (=higher secondary) or medical course but basic general knowledge needs to be included from the beginning. Male and female both should know that this is the natural process, and it should be included in health education from school level." (FDG/p12)*

**3.7.3. Reform policies and protocols for CS.** In this strategy, ten sub-themes were identified: (1) policies for avoiding CS for non-medical reasons and primigravidae/primary CS; (2) appropriate use of partograph for labour monitoring; (3) fixed service charge for CS; (4) rewards for, and investigations into hospitals in relation to CS rate; (5) monitoring of private hospitals; (6) use of Robson classification to enable proper comparison; (7) VBAC/trial of labour; (8) decision making around CS to be made by two consultants; (9) provision of security of service providers; and (10) commitment to implementation of international recommendations (including midwifery model of care and respectful maternity care) (S2 Fig).

Most interviewees stressed that policy should be reformed to avoid conducting CS on non-medical grounds. CS must only be done to avoid mortality and/or significant morbidity. Some interviewees also highlighted that primary CS should be avoided wherever possible:

*"…you should not perform the operation during the first delivery, because if the CS was performed at the primigravida, if VBAC was not performed, then again, the caesarean section will be performed at second time." (K4)*

Furthermore, many stressed that hospitals should avoid doing CS on request of the mother:

*"In private institutions we should reduce maternal requests…" (P11)*

Some emphasized that policy for reducing unnecessary CS such as CS for non-medical reasons and primary CS should be reformed and strictly monitor it.

*"The first step to reducing unwanted CS is to make strict protocol… Secondly, strict monitoring of whether the protocol has been strictly followed or not. The main thing is to reduce the unwanted CS, one had to make a protocol and strict monitoring of it…The current policy has everything, but we do not have a system of strict monitoring. If the government alone can't monitor, it is better to have a separate body…In addition, there is policy which need to be strictly implemented, there is system to provide allowances to the mother who give birth through normal delivery." (K1)*

Compulsory use of a partograph to monitor labour in each hospital was emphasised as a strategy. Some interviewees stressed that labour in hospital should be monitored around their use of the partograph, although there was a notion that some private hospitals did not use them:

*"Our hospital has its own protocol. We complete partograph but I don't think private hospital use partograph, at least partograph should be filled." (P6)*

Some interviewees recommended that the government should create policy to reward hospitals with high normal birth, to investigate reasons for high CS rates, to run programmes to increase morale of doctors and to promote audits to reduce unnecessary CS. Such policy reform could encourage more normal deliveries in hospitals:

*"…create protocol/guideline to be followed to reduce unwanted CS... programs to increase morale of the physicians, encouraging strategy. The hospital that has more normal delivery had to be awarded. If it is encouraging there will be competition in the hospital, so more vaginal delivery can be done or in a hospital where most of the CS are done, the government should investigate the cause of more CS. Auditing can also reduce CS because when there is inquiry, then people become aware that unnecessary CS should not be done…" (P10)*

Several stressed that the Government of Nepal should monitor private hospitals regarding high CS rates as CS should only be conducted based on indications:

*"Government should advocate that CS should be done only in indicated cases… Government should also monitor the private sector and need to strengthen for doing CS in indicated cases only." (K5)*

Some interviewees highlighted that the government should not blame private hospitals because doing a CS is an easy option to generate income for these hospitals:

*"Private hospitals are there for making money unless and until government sector they do not themselves question and monitor them, nothing is going to happen. That is why it is useless to blame private hospital, because we are pretty much sure they are for profit making and they don't get any money from government to run hospital and to run hospital they must generate fund, and to generate the fund CS is best option…" (K3)*

Some interviewees emphasised that fees for CS must be fixed in private hospitals. The current fee for CS in the private hospitals is high but also varies across hospitals. Poor families cannot afford CS in private hospitals. Therefore, the government should set a fixed CS rate at a reasonable price and should monitor it:

*"… financial aspects like in government sector CS is free, likewise if there are certain rules like only CS service fee can be charged up to that amount this will help patient to get service easily. The fee in a private hospital is very expensive for CS…It is cheaper in some places. Doing CS for some of our poor family or lower/middle-class family is expensive. So, even if they need to do CS, they try for normal delivery. At the same time, the lives of both mother and baby may be at risk, so if the Government of Nepal makes fixed rate available to the public at a very reasonable price along with strict monitoring... those who need a CS will get CS." (K1)*

A few interviewees suggested that the Robson criteria must be applied in all hospitals. This would enable better identification of the reasons for CS and allows cross comparison locally, regionally and internationally:

*"By taking the Robson protocol and analysing all the hospitals indication for CS. This should be applied in private as well. Going private is the individual choice of the patient but Robson criteria is universally accepted and approved by WHO. Now there is also a necessary to know why the CS rate is high. The root cause also comes from the criteria and if the main cause of CS is found then we can reduce the unwanted CS." (P9)*

Many interviewees discussed the provision of protocols for VBAC and trial of labour for breech presentation. It would not only reduce CS rates, but also reduce the cost:

*"…we should give a chance of trial of labour to previous CS, and this must be kept in every hospital protocol. Usually, doctors are distressed while the baby takes long time to deliver in order to be on the safe side, we want to do CS. So, it would be better if doctors try to keep calm and manage their distress and not perform unnecessary CS." (P11)*

One interviewee highlighted the need for intensive monitoring in a trial of labour in breech presentation to reduce CS:

*"…it would be better if hospital could provide intensive monitoring for the trial of labour in the case of breech and previous CS. If we do all this, then normal physiological birth can be promoted." (P9)*

Many interviewees highlighted the provision of a safe working environment by protecting staff's safety and morale to help reduce defensive CS:

*"…Providers also should have confidence and safety. In case of any problem during delivery, safety of the provider should be ensured. The society should also be aware if anything happens blaming doctors, attacking them should not be done. Government should provide guarantee of security and conduct different workshops to increase the morale… This is the responsibility of the hospital as well as of government to provide safe environment for health worker to work." (P10)*

Some interviewees also emphasised the need for the law to protect health workers if they followed protocol even in case of bad outcomes as patients' family are often seeking revenge:

*"There should be a provision for safety of the doctor. If the system has followed well there, the judge does not have to make a hasty decision. If you have not followed the system, you must be punished if you have made a mistake, but even after following the system, sometimes bad results come, it does not mean that we know everything about the child inside... There should be strict law for the sake of doctors in what condition do they get exemption and punishment if any complications arise even if precaution is taken as protocol." (K4)*

Implementing international maternity recommendations was also mentioned as a strategy by interviewees. Some emphasised that government policy should aim to reduce CS by following the WHO protocol on the provision of Respectful Maternity Care with the midwifery model of care:

*"The maternity department head is obstetrician, he/she should commit first to reduce unnecessary CS, they should commit that their institution CS rate should not be more than WHO standard... Department of Health Services, Family Welfare Division Head all should be committed collaboratively to reduce unnecessary CS… from national level WHO criteria should be strictly followed. Similarly, local level government should also commit for respectful maternity care... The Ministry of Health should be conceptually clear which model exists for maternity care in the country... This should be applied in the private sector as well." (K3)*

**3.7.4. Promotion of physiological birth.**  In this strategy, eight sub-themes were explored: (1) promote midwifery model of maternity care; (2) manage low-risk cases by midwives in birthing centres; (3) provide VBAC and trial of labour; (4) promote instrumental delivery; (5)

provide painless delivery; (6) involve husbands; (7) improve care during pregnancy; and (8) improve care in public hospitals (S2 Fig).

Most interviewees emphasised that physiological birth should be promoted by employing midwives to avoid certain interventions, such as using oxytocin unnecessarily.

> *"…the approach to delivery should be midwifery model by promoting physiological birth, then the rate of caesarean section might be lowered... Because midwives have more knowledge about physiological birth and how to perform normal delivery. And the way in which the oxytocin is being used randomly and the misoprostol an induction is increasing. After the induction fails, the next option is caesarean section. So, we have to promote physiological birth giving process by producing more midwives and utilising them..." (P1)*

There is a birthing centre in the public hospital where low-risk cases are managed by SBA nurses and midwives. Similar birthing centres should be established in other hospitals. Some interviewees emphasised that low-risk cases should be looked after by midwives and trained nurses in birthing centres to promote physiological birth, as in their maternity hospital:

> *"…should promote the system of delivery like in our maternity hospital, low risk cases are there in birthing centre where nurses and midwives manage delivery. The caesarean section rate over there is less than 10%. Therefore, birthing centres should be established in all the places in the country where there are more than 5000 deliveries in a year, but that birthing centre should not be like which is there in the health post. There should be a birthing centre having obstetric unit where nurses, midwives can do normal physiological delivery…" (K3)*

Moreover, intensive monitoring of labour for VBAC and trial of labour for breech presentation can promote physiological birth:

> *"…hospital could provide intensive monitoring for the trial of labour in the case of previous CS and breech presentation. If we do all this, then normal physiological can be promoted." (P9)*

Some interviewees suggested that more instrumental deliveries could reduce CS rates:

> *"We also do vacuum, forceps, instrument delivery we need to keep both mother and baby healthy…instrumental/vacuum delivery should be performed like previously if these things are done then we can reduce CS rate." (P6)*

Several indicated that provision of pain relief in labour can help reduce maternal request CS. However, there is still a challenge due to shortage of staff such as anaesthetists:

> *"…if epidural anaesthesia is provided to relieve the labour of pain. then normal delivery can be increased. Many patients are demanding painless delivery but to get painless delivery there should be good backup of anaesthetic doctors because pain management is not the scope of work of the obstetrician. However, there is a shortage of anaesthesia doctors in Nepal. Therefore, if the doctor of anaesthesia can produce and utilise it well, and then, painless delivery can be done in Nepal." (K4)*

Some interviewees emphasised that husbands should be involved during pregnancy and childbirth, to help them understand when and how to encourage partners to go to hospital in good time:

*"Husbands should be involved in the time of birth as they can't do anything during pregnancy... in our country they don't have any clue about that as they know only if their wife is pregnant. So, I think change needs to start from our society as well." (FGD/p12)*

Quality care during pregnancy can have a huge influence on the pregnant woman. Interviewees emphasised the importance of such care, including regular exercise, mental preparation and family support in promoting physiological birth.

*"…Women should be mentally strong enough to give normal delivery. Normal exercise should be done regularly from the beginning. Nowadays, there is a lack of exercise among pregnant women. They start bed resting from 2-3 months of pregnancy. Resting in bed only will make body stiff and weak. So, regular exercise is important." (FGD/P22)*

Similarly, emotional and mental support from family was important

*"…Family support for normal delivery. Pregnant mother should be mentally prepared. … Emotional and mental support from family as well. Positive thoughts in mother." (FGD/P18)*

Some referred to quality of care in public hospitals other mentioned attitudes of staff:

*"Hospital should be more concerned about the patient. Behaviour of nurses in public hospital should be good. Stiches should be done properly in the public hospitals in case of normal delivery. Before giving baby to the mother baby should be examined thoroughly and should be given to the parents." (FGD/P21)*

Moreover: *"…Health workers should speak calmly and softly without harsh words…"* (FGD/6)

## 4. Discussion

### 4.1. CS rate and indications

Overall, the CS rate was high at 50.2%, and nearly twice as high in the private hospital (68.5%) than in the public hospital (37.1%). The rate of CS in the private hospital was higher than reported previously in same hospital in 2019 [18], but so was the CS rate in the public hospital [19–21].

Previous CS (27.4%) was overall the most common indication for any CS, elective CS and multigravida in this study. Our qualitative data support this, as do several other studies [19,22–26]. However, a systematic review reported that previous CS is the second most common indication after foetal distress [27]. The progression position of previous CS as the most common indication for overall CS in this current study could be due to the rising number of women with a history of previous CS, stressing the need to avoid unnecessary primary CS. Foetal distress was the second most common indication for overall CS [19,23,25,28,29] and the most common indication in primigravida and in the public hospital [27]. It could be due to lack of trained staff for appropriate monitoring of labour and timely management of labour, or simply not monitoring the foetal heart rate appropriately [30].

Although a minority, some CSs are conducted for non-medical indications such as maternal request and not specified reasons as reported elsewhere [9,27]. Severe of note, maternal request was recorded only in private hospital.

Having the cord around the baby's neck was an indication for 1.2% of the CS rate, and only in private hospital and as emergency CS. This proportion was lower than 5.5% reported

in another study in Nepal [31]. However, the qualitative results suggested maternal requests might be higher if cord round neck is identified.

## 4.2. Robson classification of CS

The rates of CS are high in low-risk Robson groups 1-4. This suggests CS are being performed in low-risk pregnancies, leading to a larger group 5 in subsequent pregnancies [32]. Lack of adequate labour management (i.e., staff) for constant labour monitoring could be a reason for performing CS in these groups. Therefore, an appropriate evidence-based approach for labour and childbirth assistance, in groups 1-4, can reduce the rate of CSs. For example, an innovative 'Project Appropriate Birth' was implemented in maternity hospitals in 2017 in Brazil, successfully decreasing the overall CS rate from 62.4% to 55.6% (10.9%) and from 49.1% to 38.6% (21.4%) in Robson groups 1- 4 by using this approach [33].

Robson group 5 was the largest contributor to the overall CS rates in this study and many studies [21,34,35]. This reflects the lack of provision for VBAC in these two hospitals in Nepal. Robson group 1 was the second highest contributor to overall CS rate, again as reported elsewhere [36,37], whilst other studies found it the leading contributor [21,38–41]. However, the risk ratio of CS was significantly lower in this group 5, as also shown in India [38].

The third contributor to overall CS rates was Robson group 2, as reported elsewhere [37,42,43] and a key contributor [28,34,44] to the overall CS rate. A study in India also showed that there was significant higher risk of CS in this group [38].

The fourth contributor to overall CS rates was Robson group 3, other studies also reported this group was a major contributor to overall CS rates [21,37,42,43,45]. However, the risk ratio of CS was found to be significantly lower in this group, as also found in India [38].

When comparing CS by Robson categories between the private and public hospital, it was evident that in the private hospital, Robson group 5 was the major contributor to overall CS rates followed by groups 1-4. Furthermore, Robson group 1& 2 had significantly higher risks of undergoing CS. In the public hospital, Robson group 1 was the largest contributor to overall CS rates, followed by groups 5, 2, 3 and 6. A scoping review showed that Robson groups 1-6 are key contributors to overall CS rates in South Asia [35]. However, the risk of CS was found to be significantly higher only in Robson group 10 in the public hospital.

## 4.3. CS decision making

In this study, the doctor was usually the primary decision maker as the decision for an elective CS takes place in both hospitals in the antenatal clinic or outpatient department. Both the doctor and the pregnant women and/or her family, agree to do a CS. Decision making around emergency CS is made by the senior doctor when complications arise during labour. The role of women or their family in decision making in emergency CS appears limited to obtaining of consent. Of course, women usually rely on the expertise of doctors and their final decision on CS [46]. Shared decision making is way forward not only to make informed decisions but also to reduce the use of CS [47].

In hospitals in Nepal, communication should be improved to include all risks and benefits of both vaginal birth and CS to a pregnant woman and her family to reach informed consent. Effective communication skills (such as listening to women about their choice of childbirth mode) should be applied and more information, avoiding medical jargon, should be offered by providing pamphlets and leaflets on mode of childbirth including indications, risks and benefits of CS and vaginal birth. Moreover, informed consent includes the right to refuse or withdraw from the procedure. Hospital staff should offer individual counselling to pregnant

women considering their preferences, circumstances, belief and knowledge. Concerns by pregnant women/her family should be addressed appropriately without being pressured.

## 4.4. Factors affecting rise of CS in urban hospitals

**4.4.1. Medical factors.** In this study, repeat CS was the key factor influencing the rising CS rate, as all women with a history of previous CS underwent a further CS. The incidence of repeat CS was equally high (30%) in Kathmandu Medical College [48]. Some have argued that the lack of resources means that VBAC may not be the best alternative in a low-resource country like Nepal [49], as there are several factors associated with successful VBAC [50]. Although the chance of uterine rupture is low [51], staff and women may perceive repeat CS as the best option. The high proportion of repeat CS reflects the idea of 'Once a caesarean section, always a caesarean section' similar to other countries [52].

CS for a breech presentation was another medical factor contributing to the rising CS. Both Robson groups 6 (primigravida breech) and 7 (multigravida breech) had a 100% CS. Management of breech presentation is still debateable. External cephalic version for breech presentation may not common in Nepal, although it can promote vaginal birth [53].

Both hospitals are tertiary care hospitals and many complicated high-risk pregnancy cases are referred from other parts of the country. These women may need CS due to pregnancy and childbirth complications for medical indications. Both in India and Bangladesh, high-risk pregnancy is a reason for high rates of CS [54,55].

Gestation was associated with mode of childbirth; gestational age of 30-36 weeks was 0.78 times less likely to have a CS compared to full-term (37-40 weeks) in this study. Similar results have been reported in other studies [28,38]. Planned elective CS (for breech presentation, previous CS, pregnancy after IUI/IVF) is usually reserved for term pregnancies which may explain why CS rates at term are higher. Also, preterm labours may be much shorter, with less time to decide about mode of birth.

This study suggests that induction of labour in low-risk pregnancy can reduce the rate of unnecessary CS as reported by other studies [28,56], however, the rate of induction was low in our sample (11.5%). The shortage of SBAs limits the use of induction of labour because constant monitoring of labour is vital [57]. In contrast, a US study reported that the CS rate is significantly higher among women who had induced labour than those with spontaneous onset of labour [58].

**4.4.2. Sociodemographic factors.** The increasing age of first-time pregnant women is a factor in the rising CS rate. Most interviewees described how modern urban Nepalese women are getting married/starting their family later, as observed elsewhere [27,28,54]. In addition, advanced maternal age also associated with reduced success of VBAC [59], complications which are associated with an increased risk of CS [60]. Moreover, CS is viewed as a prestigious delivery for educated, wealthy urban women [61], as in Bangladesh [54,62], and elsewhere in South Asia [27]. The study showed that women from higher caste had significantly higher odds (OR: 1.14) of CS, as was found in India [63]. Whilst so-called 'precious baby', those conceived after long-term subfertility or infertility treatment, are more likely to end up as a CS [27,64].

Legislation on violence against the healthcare providers was passed in 2022 in Nepal to provide a safe working environment for all health workers [65,66]. However, violent repercussions and demands for compensation still exist in Nepal [67] when a vaginal birth goes wrong. This situation has resulted in doctors carrying out defensive CS due to fear of physical assaults and litigation [9], and a similar situation exists in Iran [61].

**4.4.3. Financial factors.** The study revealed that conducting CS is a lucrative income-stream for non-governmental or private hospitals. CS is a major operation so a high price

can be charged, not only for the operation but also for the hospital bed, medications and service charges. Those who can afford the cost and are keen to give birth by CS often go to private hospitals [68]. Financial gain can lead to private hospitals conduct more CS [9,54,61], fulfilling patients' demands [69]. Whilst in public hospitals in Nepal are receiving financial incentives from the government to enable them to conducting free CS [70], perhaps the higher rate of incentive for a CS encourages doctors to conduct more CS in public hospitals.

**4.4.4. Non-medical factors.** Only 2.7% of all CSs were conducted for maternal request, however, these only occurred in the private hospital, representing 4.8% of all is births. It may reflect that private hospitals would accept the maternal request for financial reasons. The proportion of CS conducted at maternal request was slightly lower than reported by an earlier study (6%) in the same hospital [18].

This is very interesting since all pregnant women in the FGDs preferred to give birth vaginally. The women understood the benefits of physiological birth, but poor ANC education and a lack of respectful maternity care possibly encourages CS. Perceiving CS as a safer option is another possible reason for opting for CS. Regular consultations with doctors antenatally and trusting in their advice, may lead to women changing their choice of mode of childbirth. The main reason for requesting CS was labour pain as reported by another study in Nepal [31] and studies elsewhere [9,54,61]. Pregnant women hear horror stories about labour pain from relatives and friends. Women may also request CS during labour due to not coping with pain [71]. The interviews revealed that there are few pain relief methods (e.g., use of nitrous oxide), whilst the provision of regional analgesia is uncommon in Nepal in many hospitals due to a lack of resources such as anaesthetists, facilities and awareness [72,73]. Moreover, an epidural seldom makes birth painless. The scoping review on non-medical reasons for CS suggested similar reasons for requesting a CS [9]. In 5.7% of cases in this study the reason for CS was not specified, similar to several South Asian countries [27]. Some doctors also perceive CS as safer [61].

**4.5.5. Health service-related factors.** Several health service-related factors contribute to rising CS rates in urban hospital in Nepal. The study showed that lack of appropriate antenatal counselling for pregnant mothers was a health service-related factor for rising rates of CS in urban hospitals in Nepal. They may have a lack of information or misinformation regarding childbirth [61] due to poor counselling antenatally on the mode of childbirth [74] or poor communication between doctors and women [46]. Inadequate knowledge (i.e., health consequences of CS) may lead to a preference for a CS [75], and an inability to make an informed choice about birth [61,76].

The study showed that the probability of undergoing CS was significantly higher among women who had four or more ANC visits (OR: 1.26; AOR: 1.26) than those with fewer visits as reported elsewhere [54,63,77]. On the one hand, more ANC might be needed due to pregnancy complications that need a CS. On the other hand, more ANC visits reflect the higher educational and socio-economic status of women; both are associated with an increased risk of CS.

Properly trained midwives who can advocate for the rights of the woman, counsel and motivate for normal birth are scarce in Nepal [78,79]. In urban hospitals, vaginal birth may not be promoted because this mode of delivery time consuming and there is insufficient staff [69]. In addition, lack of skills in instrumental and vaginal breech birth adds to higher CS rates [80]. Tertiary and private hospitals offering comprehensive obstetric care are mostly centralised in cities [81], and women who prefer a CS may self-refer to these [82]. The probability of having a CS was higher in private and charitable facilities than in public facilities in Nepal [83].

A written protocol for CS (currently lacking in both hospitals), following international guidelines, could encourage evidence-based practice as would the reliable use of the Robson categories which force recording of reasons for each CS and therefore enable proper audit [84].

The current adherence to the obstetric model of care in hospital and a lack of policy promoting maternity care is adding to the rise in CS. Political commitment with evidence-based international guidelines to the midwifery model of maternity care would be a vital not only for the development of the midwifery profession in Nepal [85], but also, for the reduction of unnecessary CS and promotion of physiological birth [86].

The lack of respectful maternity care is also a reason for unnecessary CS in Nepal, the preferences and choices of pregnant women may be ignored by health professionals [87]. Respectful maternity care is recommended by the WHO [88], but disrespectful maternity care was witnessed in hospitals in Nepal [89]. Therefore, lack of respectful maternity care and disrespectful maternity care would encourage women to either request CS or accept advice from a doctor and opt for CS despite them preferring vaginal birth. [61]. In addition, there is no monitoring system of the CS rate by the government as private hospitals run almost autonomously, this creates a lack of accountability for private hospitals [90].

The study emphasised that a system for auditing CS is lacking. A systematic review highlighted that audit and feedback using the Robson classification can reduce CS rates [91]. However, it seems to be far behind in auditing CS by using Robson classification in hospitals in Nepal.

## 4.2. Strategies to reduce high CS rate in urban hospitals

Evidence suggest that locally tailored multifaceted strategies are required to reduce CS use and increase physiological birth for healthy women and babies [92]. First, the provision of adequate numbers of SBAs or midwives should be a vital strategy. The Government of Nepal has been providing SBA training [93]. This training should be mandatory for all new doctors and nurses as part of continuous professional development (CPD) training. An adequate number of trained SBAs should help reduce the CS rate and promote normal birth [94]. Secondly, this study highlighted the importance of education and utilisation of midwives. Midwifery education has already been introduced in Nepal [93,95], however, there are many challenges in building an autonomous midwifery profession [85]. Thirdly, hospitals, including private hospitals, should also be obliged to establish a birthing centre where SBAs and midwives can provide the midwifery model of maternity care to reduce CS rates.

The study highlighted that antenatal health education and counselling can play a vital role in reducing the CS rate. Pregnant women should be informed clearly about the risks, benefits, indications and contraindications for CS and the benefits and risks of normal delivery. The WHO states that antenatal education is vital in reducing unnecessary CS [96], it can help reduce fear of childbirth and increase natural birth [97–101]. Moreover, educational intervention on the mode of childbirth for husbands and wives together can encourage natural birth [102]. An individual and motivating approach of midwives promotes the empowerment of pregnant women and belief in their capability to handle the birth [103]. Pregnant mothers should be well informed about mode of childbirth to be empowered to make informed choices [104].

This study emphasised that the Government of Nepal must formulate and implement robust rules and regulations for private hospital to control the use of CS and to make them accountable. Likewise, decision making for CS to be made by two consultant doctors was suggested some interviewees to reduce unnecessary CS. Evidence suggested that implementing a mandatory second opinion policy with evidence-based guidelines can reduce rates of

CS [105,106]. However, decision-making on CS by two consultants may not be practical in all hospitals due to the shortage of human resources.

In terms of workplace security, a zero-tolerance policy toward workplace violence from service users can be the best protective intervention in hospitals in Nepal [107]. Hospitals should consider installing video surveillance, extra lighting, and alarm systems and minimising access by outsiders through identification badges, electronic keys and guards.

The study emphasised that a midwifery model of maternity care should be adopted to promote physiological birth in hospitals in Nepal for the reduction of CS rates. The midwifery, or social, model of maternity care is a women-centred holistic approach to enhance natural physiological processes, which optimise the health of mother and baby and results in individual mothers' satisfaction [108]. Midwives play a key role in helping to promote a natural birth and positive birth experience. Women feel an inner strength after receiving guidance, coaching and support through the birth process by a midwife which give those women a strong motivation and encouragement to continue the natural birth process up to the most intense labour [103]. All pregnant women require help from a midwife but only a few need medical help from a doctor [109]. Low-risk pregnancy should be looked after by midwives and SBA-trained nurses in birthing centres to promote physiological birth. Normal physiological birth among low-risk women is found to be higher in midwifery-led birth settings in both primiparous and multiparous women [110]. Therefore, hospitals of Nepal should adopt midwife-led maternity care models [111].

Similarly, this study emphasised that the opportunity of VBAC must be provided in all hospitals, as success rates of VBAC have been reported to be between 60-80% [50,112–114]. However, several factors are associated with successful VBAC. It is only a safe option and successful if individual risk assessment is carried out prior to the VBAC [50,113,114]. This study also highlighted that the opportunity of trial of labour must be provided for breech. It also has training and skills implications insted of issues.

Provision of pain relief during childbirth could be an option to promote vaginal birth and to avoid CS on maternal request. Women's awareness of painless labour is low [73], and analgesia for labour pain (painless birth) is not widely offered by hospitals in Nepal. Currently, there is facility for labour analgesia service for women who are willing to have painless birth in PMWH, but this facility offers this service only to two women at a time [115]. Pregnant women are also not well informed about painless birth [72,73].

The study suggested that the involvement of husbands throughout pregnancy and childbirth can be a good strategy to promote physiological birth. Despite having positive desire and potential benefits, involvement of the husbands in wives' maternity care (pregnancy and childbirth) is not recognised in Nepal due to many factors such as their availability, cultural beliefs, and traditions [116]. Couple-based antenatal education intervention is effective in increasing spousal support and decreasing elective CS rates [117]. Therefore, the health system in Nepal should focus on the inclusion of husbands in maternity care. Care during pregnancy such as regular exercise and mental preparation for normal birth and family support for normal birth could develop pregnant women's confidence to undergo vaginal birth. Furthermore, this study also highlighted that the quality of maternity care particularly in public hospitals should be improved in terms of respectful, dignified and person-centre maternity care.

## 5. Conclusion

The rate of CS is extremely high, particularly in the private hospital, and elective CS rates were higher than emergency (particularly in the private hospital). Most recent hospital data showed that CS rate is increasing high (41.8% in PMWH in fiscal year 2023/24) [118]. High CS rates reflect the medicalisation of childbirth. The emergency life-saving procedure CS is

used frequently as an alternative mode for childbirth, which is an alarming public health problem, and it must be addressed urgently. Decision making around CS is usually made by the senior doctor and women and family are only involved for obtaining consent for emergency CS. Previous CS was the leading medical indication for performing CS and maternal request and not specified reasons were non-medical indications for CS. Robson group 5 was the major contributor to overall CS rate followed by group 1, 2 and 3. CS were often conducted in low-risk Robson groups (1-4). The risk of undergoing CS was higher in Robson group 2, 5, 6 and 9. Several associated factors with mode of childbirth are identified; the odds of CS were significantly higher in private hospital, women aged 25 and over, four or more antenatal clinic visits, breech presentation, urban residency, high caste, gestational age 37-40 weeks, spontaneous labour and no labour. The qualitative analysis identified five key themes (medical factors, sociodemographic factors, financial factors, non-medical factors and health service-related factors) affecting rising CS rates. Similarly, four main strategies were identified to stem the rise of CS: (1) provision of adequate resources (SBAs, midwives and birthing centres); (2) raising awareness on mode of childbirth (antenatal education and counselling); (3) reforming CS policies/protocols on CS; and (4) promoting physiological birth.

This mixed-methods study has highlighted many factors affecting to rising CS rates and offers details of factors fuelling the rise of CS in Nepal. The reasons behind the rising CS rates in urban hospitals are multifactorial. At the same time, the study has also recommended several possible strategies to make reasonable use of CS. Collective or collaborative multiple strategies are required for reduction of CS rates. The findings would be the important evidence not only for researchers but also for policy makers to reform policies regarding rational use of CS in Nepal. Provision of quality obstetric care can reduce unnecessary CS, and it must include social support during labour, appropriate labour monitoring, analgesic medication during labour and counselling or education of pregnant women around mode of childbirth, including indications, risks and benefits of CS during antenatal visits. Similarly, evidence-based practice and guidelines must be followed.

## 6. Limitation of the study

This study included two hospitals from Kathmandu, Nepal. Though they represent both private and public sectors, these findings may not be generalised to other hospitals in the country. The medical records of the participating hospitals did not include all demographic information such as education or the occupation of the woman and her husband thus further analysis considering these variables could not be performed.

Furthermore, there was a challenge in qualitative data collection due to many reasons such as the busy work schedules of interviewees, not enough room for conducting interview and FGDs.

## Supporting information

**S1 Table. Sociodemographic characteristics of women.**
(PDF)

**S2 Table. Indications of CS in two hospitals (public & private).**
(PDF)

**S3 Table. Indications of CS in elective and emergency CS.**
(PDF)

**S4 Table. Indications of CS in Nullipara and Multipara.**
(PDF)

**S5 Table. Robson ten group classification contribution to overall CS rate.**
(PDF)

**S6 Table. Contribution of Robson groups to overall CS rate in two hospitals** .
(PDF)

**S7 Table. Robson ten group classification and risk ratio in study sample** .
(PDF)

**S8 Table. Risk of CS in Robson groups in two hospitals.**
(PDF)

**S9 Table. Association between mode of childbirth and sociodemographic/obstetrics characteristics of participants.**
(PDF)

**S10 Table. Bivariate/multivariate logistic regression selected variables by mode of childbirth.**
(PDF)

**S1 Fig. Factors contributing to rising CS rates in urban hospitals in Nepal.**
(PDF)

**S2 Fig. Strategies for optimal use of CS in urban hospitals in Nepal.**
(PDF)

## Acknowledgments

The authors would like to thank the two hospitals in Nepal and all participants. They also thank Caspian Dugale, Dr. Astha Dhakal, Suagya Bhusal and Sudip Khanal for their support.

## Author contributions

**Conceptualization:** Sulochana Dhakal Rai, Edwin van Teijlingen, Pramod R. Regmi, Juliet Wood, Ganesh Dangal, Keshar Bahadur Dhakal.

**Data curation:** Sulochana Dhakal Rai, Edwin van Teijlingen, Pramod R. Regmi, Juliet Wood, Ganesh Dangal, Keshar Bahadur Dhakal.

**Formal analysis:** Sulochana Dhakal Rai, Edwin van Teijlingen, Pramod R. Regmi, Juliet Wood, Ganesh Dangal, Keshar Bahadur Dhakal.

**Funding acquisition:** Sulochana Dhakal Rai, Keshar Bahadur Dhakal.

**Investigation:** Sulochana Dhakal Rai, Edwin van Teijlingen, Pramod R. Regmi, Ganesh Dangal.

**Methodology:** Sulochana Dhakal Rai, Edwin van Teijlingen, Pramod R. Regmi, Juliet Wood, Ganesh Dangal, Keshar Bahadur Dhakal.

**Project administration:** Sulochana Dhakal Rai, Ganesh Dangal.

**Resources:** Sulochana Dhakal Rai, Edwin van Teijlingen, Pramod R. Regmi, Juliet Wood, Ganesh Dangal, Keshar Bahadur Dhakal.

**Software:** Sulochana Dhakal Rai.

**Supervision:** Edwin van Teijlingen, Pramod R. Regmi, Juliet Wood, Ganesh Dangal.

**Validation:** Pramod R. Regmi, Juliet Wood, Ganesh Dangal.

**Writing – original draft:** Sulochana Dhakal Rai.

**Writing – review & editing:** Edwin van Teijlingen, Pramod R. Regmi, Juliet Wood, Ganesh Dangal, Keshar Bahadur Dhakal.

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
