## [Decision Letter · Decision Letter 0]

8 Nov 2024

PONE-D-24-07372Explaining rising caesarean section rates in urban Nepal: A mixed-methods studyPLOS ONE

Dear Dr. Dhakal Rai,

Thank you for submitting your manuscript to PLOS ONE. After careful consideration, we feel that it has merit but does not fully meet PLOS ONE’s publication criteria as it currently stands. Therefore, we invite you to submit a revised version of the manuscript that addresses the points raised during the review process.

We look forward to receiving your revised manuscript.

Kind regards,

Hlengani Lawrence Chauke, MBCHB, BTh, Dip HIV Man, FCOG, MMED (O &G), MSc

Academic Editor

PLOS ONE

Journal Requirements:

2. In the online submission form, you indicated that [Data are available from first author].

5. Please upload a copy of Figures 1 and 2 to which you refer in your text on pages 15 and 20. If the figure is no longer to be included as part of the submission please remove all reference to it within the text.

6. We notice that your supplementary figures are included in the manuscript file. Please remove them and upload them with the file type 'Supporting Information'. Please ensure that each Supporting Information file has a legend listed in the manuscript after the references list.

Additional Editor Comments :

Article : PONE-D-24-07372

Title: Explaining rising caesarean section rates in urban Nepal: A mixed-methods study

Summary

Given the rising rate of caesarean sections in Nepal, this concurrent cross-sectional mixed-method study was conducted in two urban hospitals (one private and one public) in the Kathmandu district in 2021 to explore the factors contributing to the increasing rates of caesarean sections and to identify strategies that could help mitigate them. The quantitative component involved a record review of 661 births (276 from the private hospital and 385 from the public hospital) that occurred in the year 2018/19. The qualitative component included interviews with 14 health professionals (doctors, nurses, and midwives), five key informants from relevant organisations, and four focus group discussions with pregnant women attending antenatal clinics at the two study sites. Quantitative data were analysed using SPSS v28, and qualitative data were thematically analysed using NVivo v12. Data integration occurred during the discussion.

The study found a CS rate of 50.2% with higher rate in the private compared to the public hospital ( 68.5% vs. 37.1%) and previous CS was the leading indication. The following factors were associated with an increased odd of caesarean section: delivering in private hospital; aged 25 years and above; having four or more antenatal clinic visits; breech presentation; urban residency; gestational age 37- 40 weeks; spontaneous labour; and women not in labour. Robson group 5 (13.9%) was the largest contributor to overall CS rate, followed 3 by group 1 (13.4%), 2 (8.8%), 3 (4.4%) and 6 (2.9%). Similarly, the risk of undergoing CS was high in Robson groups 2, 5, 6 and 7.

Five themes related to the contributing factors for the increasing caesarean section (CS) rate and four strategies to mitigate this rise emerged from the qualitative component of the study. The five contributing factors identified were: (1) medical factors (repeated CS, complicated referral cases, and breech presentation); (2) sociodemographic factors (advanced maternal age, previous baby, and defensive CS); (3) financial factors (income for private hospitals); (4) non-medical factors (maternal request); and (5) health service-related factors (lack of awareness among midwives/resources, urban centralised health facilities, and absence of appropriate policies and protocols). The four main strategies identified by the study participants to reduce the rising CS rate were: (1) provision of adequate resources to support care during labour and birth (midwives/trained staff and birthing centres); (2) raising awareness of the risks and benefits associated with delivery methods (antenatal education/counselling and public awareness campaigns); (3) reforming CS policies and protocols; and (4) promoting physiological birth.

The authors concluded that the high Caesarean section rate in the private hospital reflected the medicalisation of childbirth, an issue of public concern that warrants urgent attention. Furthermore, the study provided insight into the contributing factors and strategies to mitigate them.

Comments

The study addresses a significant global issue, namely the rising rates of caesarean sections, due to their association with maternal morbidity and mortality. The manuscript is scientifically sound. The introduction provides a comprehensive background to the study, is well-crafted, and identifies the existing gap in the literature. The study design is appropriately aligned with the study's objectives. The methods section is clearly articulated, and there is coherence between the data analysis and the study's aims. The results section is clear and easy to follow. The discussion and conclusion sections are suitably developed. Relevant references have been appropriately cited.

The following comments are made to further improve the manuscript:

1. Grammar : The manuscript will benefit from English language editing to improve flow and reduce repetition

2. It would be beneficial to the readers if you provide additional background on the number of public and private hospitals in Nepal , the area where the study was conducted and how or why the two hospitals were selected.

3. About sample size calculation and error margin of 10% was added to the private hospital sample to compensate for incomplete records which was not the case with public hospital sample. Does this mean that public hospital records in Nepal are always complete? This seems not to be the case in most of the studies around the world. Suggest you provide an explanation regarding this decision.

4. Please include a statement on the sampling strategy for the interviews. Did you use convenient sampling technique?

5. Please provide a comment on the type of thematic analysis used in this study.

6. Results- please revise first paragraph to remove some redundancy , e.g. The overall CS rate was 50.2% with higher rate in the private (68.5%) compared to the public (37.1%) and higher rate of elective CS (55.7%, 61.9% ) compared to emergency (44.3% ,38.1%) in the public and private hospital respectively. Consider using the same approach to the rest of this section (focus on key findings and avoid repetition).

7. Please also respond to the comments from the reviewers

Thank you once more for submitting your study to PLOSONE. Looking forward to the revised manuscript.

Reviewers' comments:

Reviewer's Responses to Questions

**Comments to the Author**

1. Is the manuscript technically sound, and do the data support the conclusions?

Reviewer #1: Yes

Reviewer #2: Yes

2. Has the statistical analysis been performed appropriately and rigorously? 

Reviewer #1: Yes

Reviewer #2: Yes

3. Have the authors made all data underlying the findings in their manuscript fully available?

Reviewer #1: Yes

Reviewer #2: Yes

4. Is the manuscript presented in an intelligible fashion and written in standard English?

Reviewer #1: Yes

Reviewer #2: Yes

5. Review Comments to the Author

Reviewer #1: The study topic is important; indeed, CS rates have increased over the years and are above the WHO recommended rates. The objectives and findings of the study are clearly defined and written in an appropriate scientific manner. The manuscript is easy to ready and follow.

However, I have few comments:

1. Under Introduction the statement below could be better phrased:

Globally, rates in 2000 were double (21·1% vs 12.1%) that in 2015 [6]. It is predicted that the global CS rate will increase to 28.5% in 2030 with the lowest rate (7.1%) in Sub-Saharan Africa and highest (63.4%) in Eastern Asia [7]

2. Under methods:

a) The study method could be better elaborated, in terms of qualitative data collection. Were the interviews collected at the same as (concurrent) with data collection from the files?

b) The selection patient's file, if it could be elaborated on how the files were chosen.

c) Briefly summarising what the qualitative data questions asked

3. Under 3.5 Factors associated with mode of childbirth:

The findings and the statements quoted from participants feels more like repetition, in some of instances bringing no additional value.

For example, below under section 3.7.3 below

3.7.3 Reform policies and protocols for CS

Some interviewees recommended that the government should create policy to reward hospitals with high normal birth, to investigate reasons for high CS rates, to run programmes to increase morale of doctors and to promote audits to reduce unnecessary CS. Such policy reform could encourage more normal deliveries in hospitals:

“…create protocol/guideline to be followed to reduce unwanted CS... programs to increase morale of the physicians, encouraging strategy. The hospital that has more normal delivery had to be awarded. If it is encouraging there will be competition in the hospital, so more 26 vaginal delivery can be done or in a hospital where most of the CS are done, the government should investigate the cause of more CS. Auditing can also reduce CS because when there is inquiry, then people become aware that unnecessary CS should not be done…” (P10)

4. Table 1 & 3 could be combined Table 4 & 6 as well

Reviewer #2: 1. Please clarify the period and duration of the study in the abstract, similar to the manuscript, as the abstract has two different study periods (2021 & 2018-19).

2. Kindly provide the most recent CS rate report.

3. As the authors have mentioned 14 healthcare professionals, please ensure that you mention how many doctors, nurses, and midwives specifically.

4. Do you have any questionnaires based on which the interview was conducted?

5. Was the questionnaire validated? If so, mention the same in the methodology section of the manuscript. If the questionnaire was not validated, mention the same as limitations of the study.

6. Was pre-test done for questionnaire validation? If so, what is the percentage of the population on which the pretest was performed?

7. Please provide a table of demographic characteristics of the patients included in the study.

8. Ensure to provide figure legends for the figures.

9. Please mention the ethical approval number in the main manuscript.

10. Discuss the limitations of the study.

6. PLOS authors have the option to publish the peer review history of their article (what does this mean? ). If published, this will include your full peer review and any attached files.

**Do you want your identity to be public for this peer review?** For information about this choice, including consent withdrawal, please see our Privacy Policy .

Reviewer #1: **Yes: ** Dr Louisa Boledi Seopela

Reviewer #2: No

---

## [Author Response · Author response to Decision Letter 0]

13 Dec 2024

We have addressed all issues that raised by reviewers and the journal. Please see the cover letter and revised manuscript.

---

## [Editor Report · Decision Letter 1]

17 Jan 2025

Explaining rising caesarean section rates in urban Nepal: A mixed-methods study

PONE-D-24-07372R1

Dear . Dhakal-Rai, Sulocahana

We’re pleased to inform you that your manuscript has been judged scientifically suitable for publication and will be formally accepted for publication once it meets all outstanding technical requirements.

Kind regards,

Hlengani Lawrence Chauke, MBCHB, BTh, Dip HIV Man, FCOG, MMED (O &G), MSc

Academic Editor

PLOS ONE

Additional Editor Comments (optional):

Thank you for the revision. I can confirm that all the issues raised have been satisfactory addressed.

---

## [Editor Report · Acceptance letter]

PONE-D-24-07372R1

PLOS ONE

Dear Dr. Dhakal Rai,

I'm pleased to inform you that your manuscript has been deemed suitable for publication in PLOS ONE. Congratulations! Your manuscript is now being handed over to our production team.

Kind regards,

on behalf of

Prof Hlengani Lawrence Chauke

Academic Editor

PLOS ONE